# Detection of BRAFV600E in Liquid Biopsy from Patients with Papillary Thyroid Cancer Is Associated with Tumor Aggressiveness and Response to Therapy

**DOI:** 10.3390/jcm9082481

**Published:** 2020-08-02

**Authors:** Kirk Jensen, Shilpa Thakur, Aneeta Patel, Maria Cecilia Mendonca-Torres, John Costello, Cristiane Jeyce Gomes-Lima, Mary Walter, Leonard Wartofsky, Kenneth Dale Burman, Athanasios Bikas, Dorina Ylli, Vasyl V. Vasko, Joanna Klubo-Gwiezdzinska

**Affiliations:** 1Department of Pediatrics, Endocrine Division, Uniformed Services University of the Health Sciences, Bethesda, MD 20814, USA; kirk.jensen@usuhs.edu (K.J.); aneeta.patel@usuhs.edu (A.P.); maria.mendonca-torres@usuhs.edu (M.C.M.-T.); john.costello.ctr@usuhs.edu (J.C.); dorina.ylli@umed.edu.al (D.Y.); vasyl.vasko.ctr@usuhs.edu (V.V.V.); 2National Institute of Diabetes and Digestive and Kidney Diseases, National Institutes of Health, Bethesda, MD 20814, USA; shilpa.thakur@nih.gov (S.T.); cjglima@gmail.com (C.J.G.-L.); waltermf@niddk.nih.gov (M.W.); 3MedStar Health Research Institute, MedStar Washington Hospital Center, Washington, DC 20010, USA; leonard.wartofsky@medstar.net (L.W.); Kenneth.D.Burman@medstar.net (K.D.B.); athanasiosbikas@gmail.com (A.B.); 4Endocrinology Division, University of Medicine, 1005 Tirana, Albania

**Keywords:** COLD-PCR, digital PCR, *BRAFV600E*, papillary thyroid cancer, liquid biopsy

## Abstract

The detection of rare mutational targets in plasma (liquid biopsy) has emerged as a promising tool for the assessment of patients with cancer. We determined the presence of cell-free DNA containing the *BRAFV600E* mutations (cf*BRAFV600E*) in plasma samples from 57 patients with papillary thyroid cancer (PTC) with somatic *BRAFV600E* mutation-positive primary tumors using microfluidic digital PCR, and co-amplification at lower denaturation temperature (COLD) PCR. Mutant cf*BRAFV600E* alleles were detected in 24/57 (42.1%) of the examined patients. The presence of cf*BRAFV600E* was significantly associated with tumor size (*p* = 0.03), multifocal patterns of growth (*p* = 0.03), the presence of extrathyroidal gross extension (*p* = 0.02) and the presence of pulmonary micrometastases (*p* = 0.04). In patients with low-, intermediate- and high-risk PTCs, cf*BRAFV600E* was detected in 4/19 (21.0%), 8/22 (36.3%) and 12/16 (75.0%) of cases, respectively. Patients with detectable *cfBRAFV600E* were characterized by a 4.68 times higher likelihood of non-excellent response to therapy, as compared to patients without detectable cf*BRAF*V600E (OR (odds ratios), 4.68; 95% CI (confidence intervals)) 1.26–17.32; *p* = 0.02). In summary, the combination of digital polymerase chain reaction (dPCR) with COLD-PCR enables the detection of *BRAFV600E* in the liquid biopsy from patients with PTCs and could prove useful for the identification of patients with PTC at an increased risk for a structurally or biochemically incomplete or indeterminate response to treatment.

## 1. Introduction

Advances in the understanding of genetic and biologic characteristics of thyroid cancer, coupled with the development of new molecular targeted therapeutics, have led to improved diagnosis and treatment [1]. Progress in the characterization of the genomic landscape of thyroid cancers was principally fostered through next-generation sequencing (NGS) methods, allowing the detection of multiple variants simultaneously, including base substitutions, insertion/deletions, gene amplifications and gene rearrangements [2]. The analysis of genomic variants revealed a high frequency of activating somatic alterations of genes encoding effectors in the mitogen-activated protein kinase (MAPK) signaling pathway, including the point mutations of *BRAF* and the *RAS* genes, as well as fusions involving the RET and NTRK1 tyrosine kinases. *BRAF* mutations have been the most frequently detected genomic alterations in papillary thyroid cancer (PTC), being primarily mutations encoding V600E substitution [2,3,4].

On average, the frequency of the *BRAFV600E* mutation is 45% in PTC patients, but its prevalence can vary from 25–82% depending on the population examined [5,6]. The presence of *BRAFV600E* mutation in PTC patients has been associated with increased tumor progression, aggressive clinicopathological features and poorer prognosis, particularly when coupled with a telomerase reverse transcriptase (TERT) promoter mutation. Specifically, adverse clinical-pathological features associated with the *BRAFV600E* mutation include the presence of lymph node metastases, extrathyroidal invasion, distant metastases and advanced cancer stages [3,7]. Moreover, the occurrence of the *BRAFV600E* mutation has been correlated with tumor recurrence and reduced or absent radioiodine (RAI) avidity, leading to the failure of RAI therapy [8]. A linear relationship has also been observed between patient age and mortality risk in PTC patients with *BRAFV600E* mutation [3,9,10]. 

The detection of the *BRAFV600E* mutation in thyroid cancer patients has been conventionally determined by fine-needle aspiration (FNA) or tissue biopsy, followed by molecular testing and/or immunohistochemistry [11,12]. Growing evidence demonstrates that blood-based liquid biopsies provide a minimally invasive alternative for identifying the cellular and molecular signatures that can be used as biomarkers to detect early-stage cancer and predict disease progression [13,14]. In this context, the examination of circulating cell-free *BRAFV600E* (cf*BRAFV600E*) in blood could represent an attractive approach for diagnosis and monitoring the response to treatment in patients with PTC.

Since the amounts of cfDNA and cfRNA in blood are low, highly sensitive techniques are required for genomic analysis in liquid biopsy. Polymerase chain reaction (PCR) coupled with direct sequencing is a standard method for the detection of mutations, but this method is not sufficiently sensitive for the detection of low levels of DNA variants in a mixture of wild type and mutant sequences. Currently, droplet digital PCR (dPCR) is considered the technique of choice for the detection of rare mutations in liquid biopsy samples [15,16,17,18]. Recent studies demonstrated that dPCR can be successfully employed for monitoring response to therapy by quantifying *BRAF* and *RAS* mutants in samples from patients with cancers [19,20,21]. Another approach applies the techniques that allow the enrichment of low abundance variants in DNA samples. Co-amplification at lower denaturation temperature-based PCR (COLD-PCR) is a modified PCR method that allows the preferential amplification of rare mutant alleles within a target amplicon [22,23,24]. The unique attribute of COLD-PCR is that the selective enrichment of low-abundance mutations within a target amplicon is achieved by exploiting a small difference in the amplicon melting temperature [25,26]. Recent studies demonstrated that COLD-PCR was 10–100 times more sensitive than standard PCR in the detection of mutant variants. Currently, various versions of COLD-PCR (full-, ice-, fast-, TT-COLD-PCR) have been developed and utilized successfully for the detection of mutated genes, including *KRAS*, *HRAS*, *NRAS*, *EGFR*, *TP53* and *BRAF* [23,24,27].

Until now, no data have demonstrated the suitability of the COLD-PCR method in the detection of a *BRAFV600E* mutation in liquid biopsy samples from thyroid cancer patients. In the current study, we investigated the utility of COLD-PCR in combination with digital PCR for the detection of *BRAFV600E* in liquid biopsy samples from the patients with PTC. The data derived were utilized to analyze the potential utility of this approach to improve the risk stratification and response to treatment in patients with thyroid cancer.

## 2. Materials and Methods

### 2.1. Subjects

This study was approved by the Uniformed Service University of the Health Sciences (USUHS), National Institutes of Diabetes, Digestive and Kidney Disease (NIDDK), (clinicaltrials.gov identifier NCT00001160) and MedStar Washington Hospital Center Institutional Review Boards, and informed consent was obtained from each patient.

This study consisted of a retrospective analysis of blood samples obtained from patients seen at NIDDK or MedStar Washington Hospital Center for the management of thyroid nodules. All patients underwent surgery (near-total or total thyroidectomy). After surgery, formalin-fixed, paraffin-embedded (FFPE) tissue samples were subjected to Ion Torrent^TM^ Oncomine^TM^ Comensive Assay v3 (OCAv3) next-generation sequencing to determine the final pathology diagnosis and the mutation status of the original tumor. Only the patients with histologically confirmed papillary thyroid cancer (PTC) harboring a *BRAFV600E* mutation (57 patients) were included in this study.

These *BRAFV600E*-positive tumors were further sub-classified into low-, intermediate- and high-risk groups, based upon the criteria of the American Thyroid Association [28]. The low-risk group included intra-thyroidal PTC tumors with complete macroscopic tumor resection, no histological evidence of extra-thyroidal extension or vascular invasion, and no clinically evident lymph node metastases or distant metastases. PTCs that were classified as intermediate-risk tumors demonstrated evidence of microscopic invasion into perithyroidal soft tissue, vascular invasion, or clinical lymph node metastases. High-risk patients had tumors with evidence of distant metastases, macroscopic invasion of the perithyroidal tissue/structures, lymph node metastases ≥ 3 cm, or incomplete tumor resection.

### 2.2. Treatment with RAI, Biochemical Testing and Assessment of Overall Response to Therapy

Radioactive iodine therapy was administered to 46/57 patients. Serum thyroglobulin (Tg) and anti-Tg antibodies (TgAb) levels were measured by the Siemens Immulite 2000 Immunoassay system. The functional sensitivity of this assay was 0.07 ng/mL (Tg), and the lowest reported values were Tg less than 0.2 ng/mL.

The overall response to therapy was evaluated at the last follow-up visit and reported as an excellent response (no evidence of structural disease on imaging, Tg on levothyroxine (LT4) < 0.2 ng/mL and no TgAb); an indeterminate response (nonspecific findings on imaging and/or Tg on LT4 ≥ 0.2–< 1 ng/mL or stimulated Tg ≥ 1–< 10 ng/mL or stable or declining TgAb titers); a biochemical incomplete response (no evidence of structural disease on imaging and Tg ≥ 1 ng/mL on LT4 or stimulated Tg ≥ 10 ng/mL or rising TgAb titers); and structural incomplete response (evidence of structural disease on imaging).

### 2.3. Extraction of Cell Free DNA (cfDNA) from Plasma Samples

Patients underwent a blood draw as part of their treatment evaluation. In total, the blood samples were collected from 57 patients with *BRAFV600E*-positive primary PTCs. Fresh blood (2 mL) samples were drawn into BD Vacutainer EDTA tubes (Cat# 366643), immediately centrifuged at 3000 rpm for 10 min at +4 °C, and the plasma fractions were transferred to a clean tube and re-centrifuged at top speed for another 10 min. The plasma then was then carefully transferred to a clean tube to avoid any possible aspiration of the cell pellet and stored at −80 °C.

The isolation of the cfDNA was performed on the KingFisher^TM^ Duo Prime particle processor (Thermo Fisher Scientific; Waltham, MA, USA) using the MagMax Cell Free DNA isolation kit (Thermo Fisher Scientific; Waltham, MA, USA; cat#A29319). The magnetic bead-based purification format enabled the processing of as little as 600 μL of plasma. For the samples where the plasma volume was less than 600 μL, PBS was added to reach 600 μL and corrected for the dilution factor to obtain a uniform 600 μL cfDNA extraction volume for the whole group. The purified cfDNA sample was eluted in 30 μL volume. The quality of the extracted cfDNA was examined using NanoDrop (Thermo Scientific, Waltham, MA, USA), and the eluates were stored at −80 °C.

### 2.4. Detection of BRAFV600 Mutation by Digital PCR

Microfluidic digital PCR analyses were performed on a QuantStudio™ 3D Digital PCR System (Life Technologies, Carlsbad, CA, USA) as per the previously published protocol [29]. The rare mutation SNP genotyping assays for *BRAFV600E* (Fluorescein—FAM) which was multiplexed with an assay for the detection of wild-type *BRAF* (2′-chloro-7′phenyl-1,4-dichloro-6-carboxy-fluorescein—VIC) were synthesized by Thermo Fisher Scientific. Reaction mixtures contained 8.75 μL of QuantStudio 3D Digital PCR Master Mix, 0.425 μL of TaqMan Assay by Design primer–probe mix, and a diluted sample of cfDNA. A non-template control of water was run for each assay, as well as positive (BCPAP–*BRAFV600E*-mutant thyroid cancer cell line) and negative (FTC133–*BRAF*-wild-type thyroid cancer cell line) sheared DNA controls. The reaction mixture was loaded onto the Quantstudio 3D digital PCR 20K chip and amplification was performed on the Proflex PCR system under the following conditions: 96 °C for 10 min, 39 cycles at 56 °C for 2 min and at 98 °C for 30 s, followed by a final extension step at 60 °C for 2 min. After amplification, the chips were imaged on the QuantStudio 3D Instrument.

The chips were analyzed by the Quantstudio 3D analysis suite cloud software, which assesses raw data and calculates the estimated concentration of the nucleic acid sequence targeted by the FAM and VIC fluorescent dye-labeled probes according to the Poisson distribution. The thresholds for FAM and VIC detection were manually set based on the results from the no-template control, and a minimum of 15,000 counts was required for analysis. The resulting data were reported in copies/μL together with the results of the data quality assessment metrics. The mutant fraction was calculated with the ratio of FAM counts (*BRAFV600E*) over FAM and VIC counts (*BRAFV600* and *BRAFWT*) and the value standardized as a percentage.

### 2.5. Improved and Complete Co-Amplification at Lower Denaturation Temperature (ICE-COLD-PCR) 

To enhance the sensitivity of the digital PCR method in the detection of *BRAFV600E* mutations, the COLD-PCR application was applied in order to amplify rare mutant copies relative to wild-type (WT). In brief, extracted nucleic acid template (DNA) was combined with a Precipio-manufactured reference sequence (RS). RS is a synthetic, single-stranded, wild-type-specific oligonucleotide reference sequence, which binds to the wild-type template and inhibits its amplification. The oligonucleotide RS contains a 3′-phosphate modification to prevent polymerase extension. The RS is slightly shorter than the length of the PCR amplicon so that it obstructs primer binding and prevents the amplification of wild-type alleles. At a critical denaturation temperature, the RS:WT duplex remains double-stranded, inhibiting the amplification of WT through thermocycling while RS:mutant duplexes are denatured and then exponentially amplified.

COLD-PCR was performed using the *BRAF* exon 15 mutation analysis kit (Precipio, NE, USA) on the Quantstudio Flex system (Thermo Fisher Scientific; Waltham, MA, USA). cfDNA from 25 pg to 1 ng were used as input in a 25 μL PCR mix including RS-oligonucleotide/PCR primer mix, and 2× Polymerase mix. Wild-type and *V600E* controls were also included. An initial denaturation step was performed for 30 s at 98 °C, followed by 45 cycles of ICE-COLD-PCR (10 s of denaturation at 98 °C; 30 s of blocker annealing at 69 °C and 30 s at 74 °C (Tc); 30 s of primer annealing at 63 °C; and 20 s of elongation at 72 °C) to enrich the mutation fraction. An additional five cycles of standard PCR amplification (10 s of denaturation at 98 °C, 10 s of annealing at 63 °C, and 20 s of elongation at 72 °C) was performed with the final extension at 72 °C for 5 min and infinite hold at 12 °C.

Enriched samples were used for the detection of *BRAFV600E* and wild-type *BRAF* alleles by real-time PCR using the Taqman Single-nucleotide polymorphism (SNP) Genotyping assay for *BRAFV600E* (Precipio, NE, USA) or for the detection of *BRAFV600E* and wild-type *BRAF* alleles using microfluidic digital PCR. For real-time PCR, 5 uL of the diluted sample (1 to 200) was added to a master mix containing the 20X Genotyping Assay mix, and the 2× GTXpress master mix. qPCR cycling conditions included an initial denaturation step for 20 s at 95 °C, followed by 40 cycles of 3 s at 95 °C, and 20 s at 60 °C, followed a final cooling step at 4 °C. The TaqMan^®^ Genotyper and real-time PCR instrument software were used for allelic discrimination. Wild-type allele and mutant alleles utilized the VIC and FAM probe, respectively. The genotype assignments were based on the ratio between the fluorescent intensities of VIC and FAM after normalization. A high VIC/FAM ratio represented wild-type alleles and a low VIC/FAM ratio corresponded to mutant alleles. For microfluidic digital PCR, 1 μL of ICE-COLD-PCR product was used and allelic discrimination was performed as described above. Positive control (*BRAFV600-mutant* BCPAP cell line), wild-type (*BRAF* wild-type FTC133 cell line) control and negative template controls (NTC) were included and runs were done in triplicate.

### 2.6. Statistical Analysis

Data were summarized by using frequencies and percentages for categorical variables and means with standard deviations (SDs) or medians with the 25–75% interquartile range (IQRs) for continuous variables, dependent on the distribution. Shapiro–Wilk’s normality test was performed to access normal distribution. Categorical variables were compared between subgroups using the Fisher exact test or the Pearson Chi-square test, when appropriate. Continuous variables were compared with Student’s t-test and with the Mann–Whitney test when not following normal distribution. Logistic regression analyses were used to assess if mutation detection was an independent predictor of response to treatment when other clinically relevant predictive variables were kept constant. The odds ratios (OR) were reported along with their 95% confidence intervals (CI). A *p*-value of ≤ 0.05 was considered to be statistically significant in the analyses. Data analysis was performed using IBM SPSS Statistics 25.0 (IBM Corp., Armonk, NY, USA).

## 3. Results:

### 3.1. Combination of Microfluidic Digital PCR with COLD-PCR Increases the Sensitivity for Detection of BRAFV600E 

We established a protocol to detect *BRAFV600E* in thyroid cancer samples by microfluidic digital PCR and described this technique in a previously published study [29]. In the present study, we applied a similar approach and in addition, used a co-amplification at COLD-PCR prior to digital PCR. Initial experiments were performed using the control DNAs (wild-type *BRAF* and 1% *BRAFV600E*), as well as the DNA extracted from a *BRAFV600E*-positive BCPAP cell line. As demonstrated in Figure 1, COLD-PCR significantly enriched the DNA template in rare mutant copies and increased the sensitivity of digital PCR for the detection of *BRAFV600E* by 100-fold. Subsequently, we used this approach for the analysis of DNA extracted from the plasma samples of patients with thyroid cancer. 

### 3.2. Clinicopathologic Characteristics of BRAFV600E Positive Thyroid Cancers

The study cohort consisted of 57 patients with *BRAFV600E*-positive PTCs, followed for a median of 27 (IQR 16.5–51) months. Of these, 40 (70%) were women, and 17 (30%) were men. The average age of patients was 46 years, ranging from 21 to 82 years. The final pathology revealed a thyroid cancer diagnosis in all patients, including 38 patients with classical PTC (CPTC), 13 with tall cell variant PTC (TCPTC), and six with follicular variant PTC (FVPTC). In one case in addition to CPTC, a micro-medullary thyroid cancer was detected.

The average tumor size was 2.4 ± 1.8 cm, with a multifocal growth pattern found in 34/57 (59.6%) cases and gross extension in 13/57 (22.8%) cases. Lymph node metastases to the central neck and lateral neck compartments were detected in 31/57 (54%) and 18/57 (31%) cases, respectively. Distant metastases were found in 12 patients (in 10 individuals in lungs and two cases in bones). There were 19 patients with low-risk thyroid cancer, 22 patients with intermediate-risk tumors, and in 16 cases the tumors were classified as high-risk PTCs.

### 3.3. Detection of cfBRAFV600E in Serum from Patients with Low-, Intermediate- and High–Risk PTCs

Microfluidic digital PCR allowed the detection of cf*BRAFV600E* in the DNA extracted from 8/57 (14%) patients. The ratios of the mutant vs. wild-type *BRAF* alleles ranged from 0.6% to 6.9%. By microfluidic dPCR, cf*BRAFV600E* alleles were found in seven patients with high-risk PTCs and one patient with intermediate-risk PTC. The combination of microfluidic dPCR with COLD-PCR allowed the detection of circulating cf*BRAFV600E* in 24/57 (42.1%) examined patients.

Clinicopathological characteristics of the cases in which circulating cf*BRAFV600E* was detected and the cases in which wild-type *BRAF* was detected are summarized and compared in Table 1.

There was no significant correlation between the presence of cf*BRAFV600E* or wild-type *BRAF* in plasma and patient gender and age at diagnosis or histological subtype of tumor. However, the detection of cf*BRAFV600E* was significantly associated with tumor size (*p* = 0.03), multifocal patterns of growth (*p* = 0.03), the presence of extrathyroidal gross extension (*p* = 0.02) and the presence of pulmonary micrometastases (*p* = 0.04).

There was no significant correlation between the presence of cf*BRAFV600E* or wild-type *BRAF* in plasma and patient gender and age at diagnosis or histological subtype of tumor. However, the detection of cf*BRAFV600E* was significantly associated with tumor size (*p* = 0.03), multifocal patterns of growth (*p* = 0.03), the presence of extrathyroidal gross extension (*p* = 0.02) and the presence of pulmonary micrometastases (*p* = 0.04).

Analysis of cfDNA from blood samples that were obtained in patients with low-, intermediate-, and high-risk PTCs revealed the presence of cf*BRAFV600E* in 4/19 (21.0%), 8/22 (36.3%) and 12/16 (75.0%) of cases, respectively. The results of the analysis are summarized in Table 2.

In the liquid biopsy from patients with low-risk tumors, cf*BRAFV600E* alleles were less frequently detected than wild-type *BRAF* (*p* = 0.026). In contrast, in the plasma from the patients with high-risk PTCs, cf*BRAFV600E* alleles were more frequently detected than wild-type *BRAF* (*p* = 0.002).

### 3.4. Detection of cfBRAFV600E in Serum Samples and Treatment Outcome

All patients underwent surgery for thyroid cancer. During the follow-up, repeat surgery for metastatic lymph nodes was reported in 12/57 patients, including seven patients with detectable *BRAFV600E* in serum and five patients with circulating wild-type *BRAF*. Treatment with RAI was administered to 46/57 patients (23 with cf*BRAFV600E* and 23 with wild-type cf*BRAF*). Multiple treatments with RAI were administered in five patients with cf*BRAFV600E* and in two patients with wild-type cf*BRAF*. The median cumulative doses in patients with *BRAFV600E* and wild-type *BRAF* were 145 mci (IQR 250 mci) and 100 mci (IQR 150 mci), respectively, *p* = 0.022.

An overall excellent response was documented in 6/24 (25%) patients with cf*BRAFV600E*, and in 23/33 (69.6%) patients with wild-type cf*BRAF* (*p* = 0.001). The analysis of treatment outcomes in patients with low-, intermediate-, and high-risk cancers is presented in Table 3.

A multivariate logistic regression model, including clinically relevant prognostic factors such as age, ATA risk stratification and cf*BRAFV600E* mutation status, revealed that the detection of cf*BRAFV600E* was an independent factor for an incomplete response to treatment, along with the baseline pathology associated with a high risk for recurrence. Table 4 shows that the odds of having an incomplete response to treatment is 4.68 times greater for patients with cf*BRAFV600E* mutation compared to the patients with wild-type *BRAF,* independently of other risk factors (OR, 4.68; 95% CI, 1.26–17.32; *p* = 0.02).

### 3.5. Detection of cfBRAFV600E in Serum Samples from Low-Risk Thyroid Cancer Patients

Low-risk papillary cancers represent the vast majority of thyroid cancers diagnosed today. There is no convincing evidence that aggressive treatment is beneficial for these patients. Therefore, studies that identify molecular markers that would predict the response to treatment in low-risk thyroid cancer patients could be useful.

An association between the presence of cf*BRAFV600E* and an efficacy of treatment was found when a low-risk group of patients was analyzed. In total, the excellent response to treatment in the low-risk group was documented in 15/19 (78.9%) patients. Among these patients, *BRAFV600E* was detected in the serum in four individuals, and only wild-type *BRAF* was found in 15 patients. An excellent response was reported in 25% (1/4) cases with *BRAFV600E*, and in 93% (14/15) of patients with wild-type *BRAF* (*p* = 0.016).

## 4. Discussion

Insights into the biological behavior of malignancies were gained by the identification of biomarkers in blood samples, taken at various time points during the course of the disease. Liquid biopsy has been shown to be a viable approach for the minimally-invasive assessment of tumor-specific mutations, and can potentially be used in both clinical and investigational applications [13,14]. Recent studies have demonstrated that liquid biopsy is a useful technique for monitoring the response to treatment in patients with medullary and radioiodine-refractory poorly differentiated thyroid cancers [30,31]. 

In this study, we examined the utility of liquid biopsy in patients with *BRAFV600E*-mutant PTC. We focused on the detection of cf*BRAFV600E*, since this is the most frequent genomic alteration found in PTC and a well established druggable target with already approved pharmacological inhibitors of the BRAF/MAPK signaling pathway. We hypothesized that the detection of *BRAFV600E* in liquid biopsy samples from patients with *BRAFV600E*-positive primary PTCs could be correlated with the aggressiveness of thyroid cancer. We also sought to determine the potential utility of this test for the delineation of treatment outcome in patients with PTCs.

The first step was the development of a sensitive assay, capable of detecting low-abundance *BRAFV600E* in the blood samples from patients with thyroid cancer. One approach for the assessment of low-abundance DNA variants involves technologies allowing the preferential enrichment of mutant DNA sequences, such as COLD-PCR [32]. COLD-PCR can be combined with other PCR techniques and evidence indicates that COLD-PCR can be used as a fundamental platform to improve the sensitivity of downstream technologies, including Sanger sequencing, digital PCR and mutation genotyping.

We developed a protocol for the analysis of cf*BRAFV600E*, where microfluidic digital PCR was combined with COLD-PCR. By performing experiments using synthetic DNAs as well as DNA extracted from thyroid cancer cell lines, we demonstrated that the enrichment of mutant alleles using COLD-PCR increases the sensitivity of microfluidic digital PCR by 100-fold, as compared to digital PCR alone. These results suggest that the combination of microfluidic digital PCR with COLD-PCR could be a suitable technique for the assessment of liquid biopsy samples from patients with thyroid cancer.

Then, we applied this approach for the analysis of *BRAFV600E* in DNA extracted from the plasma of patients with PTC. We demonstrated that microfluidic digital PCR allowed the detection of cf*BRAFV600E* in 14% of patients. However, when the enrichment of low-abundance mutations by COLD-PCR was performed prior to dPCR, cf*BRAFV600E* was detected in the blood of 42% of the examined patients. These results show that differences in methodology significantly affect the outcome of the assay, providing an explanation for the conflicting results that have been published. The analysis of *BRAFV600E* in the serum samples from 94 patients with PTC that were performed using real-time PCR demonstrated that none of the patients had a detectable serum *BRAFV600E* mutation [33]. However, another study demonstrated that cf*BRAFV600E* was detected in 20 of 173 PTC patients when the allele-specific real-time PCR method was employed [34]. Methodological differences can also partially account for the unclear association between cf*BRAFV600E* and clinicopathologic characteristics. Thus, while there was no correlation between cf*BRAFV600E* and clinicopathologic data in one study [33], the detection of *BRAFV600E* in the blood was found to be associated with the *BRAF* status in primary tumors and the presence of active disease in another study [34].

In the current study, by using a combination of microfluidic digital PCR and COLD-PCR techniques, we showed that the detection of cf*BRAFV600E* in liquid biopsy has a lower performance as compared to *BRAFV600E* testing in thyroid tissue samples. These results are consistent with the previously published analyses of cf*BRAFV600E* in liquid biopsy samples from patients with thyroid tumors [34,35,36]. These findings suggest that molecular testing from liquid biopsy samples in patients with thyroid nodules cannot substitute for fine-needle aspiration biopsy.

In patients with thyroid nodules, the detection of *BRAFV600E* in thyroid fine-needle aspiration biopsy is a well established marker of malignancy, and is associated with a greater likelihood of nodal recurrence [5,12]. However, it appears that the *BRAFV600E* mutation alone is of limited value in risk stratification and its detection in fine needle aspiration biopsy (FNAB) samples does not provide definitive information on the degree of tumor aggressiveness, or the presence of extrathyroidal extension or of distant metastases.

In our study, a comparison of two groups of patients (with- and without detectable cf*BRAFV600E* in plasma) revealed that the presence of cf*BRAFV600E* was associated with the size of the primary tumor as well as with the presence of extra-thyroidal gross extension at the time of surgery. In addition, pulmonary micro-metastases were more frequently detected in patients with detectable cf*BRAFV600E*. These observations suggest that in patients with thyroid nodules, the diagnostic workflow might include not only the detection of *BRAFV600E* in FNAB samples, but also the assessment of plasma cf*BRAFV600E*. The combination of a solid and a liquid biopsy could potentially improve the risk stratification in patients with PTC.

The assessment of liquid biopsy samples in patients with low-, intermediate-, and high-risk tumors demonstrated a progressive increase in the frequency of detectable cf*BRAFV600E*. These observations were in agreement with previous reports that have demonstrated an association between the presence of circulating cf*BRAFV600E* and tumor burden in patients with PTC [34,35]. However, in our study, the detection of circulating cf*BRAFV600E* was not limited to the patients with advanced PTC. Moreover, we demonstrated that the analysis of liquid biopsy can help in the identification of a subset of PTCs with more aggressive characteristics even in the group of low-risk cancer patients. The use of an effective method in the early identification of a thyroid tumor could permit a minimal invasive surgery preventing some of the complication related to conventional surgery [37,38]. Our data suggest the potential utility of liquid biopsy in guiding the surgical decision, however, additional studies addressing this specific question are needed. 

Approximately 5% of metastatic thyroid cancer becomes less well differentiated and demonstrates a poor response to treatment [1]. Patients with *BRAFV600E*-positive thyroid cancer are poor responders or refractory to RAI therapy because this gain-of-function mutation modulates iodine metabolism resulting in a decreased ability of neoplastic cells to uptake and incorporate radioiodine [7,10]. Recently, the FDA approved two small BRAF-specific inhibitors: vemurafenib and dabrafenib for *BRAFV600E*-positive advanced RAI-refractory thyroid cancer and metastatic PTC. Evidence suggests that BRAF inhibitory agents restore RAI uptake in *BRAFV600E* iodine-refractory thyroid cancer cells, probably by reactivating the expression of thyroid-specific genes involved in iodine metabolism [39]. In this context, the identification of patients who would benefit from therapy with the pharmacological inhibitors of BRAF is an important issue. In this study, we demonstrated that the detection of cf*BRFV600E* in liquid biopsy is an independent factor predicting the response to therapy in patients with PTC. Further study could determine whether the evaluation of cf*BRFV600E* in liquid biopsy samples could identify patients with a risk of incomplete response after RAI treatment who might then be candidates for alternative treatment with BRAF inhibitors.

The utility of liquid biopsy for the monitoring of patients’ responses to treatment has been demonstrated in patients with melanoma, colon cancer, and medullary thyroid cancer [13,16,30]. Unfortunately, the design of the current study (retrospective analysis of plasma samples) did not allow for the evaluation of the utility of liquid biopsy for monitoring the response to treatment with RAI over time in the same patients. The only serum marker currently used in the follow-up of PTC is thyroglobulin (Tg), which is extremely sensitive and reliable but may be less informative in some subsets of patients, particularly those with anti-Tg antibodies. The incomplete resection of normal thyroid tissue or the dedifferentiation of thyroid tumors can render the interpretation of thyroglobulin extremely challenging. In such cases, circulating *BRAFV600E* may provide an additional tool to monitor PTC patients during treatment.

The development of a comprehensive set of molecular variables to improve risk stratification and facilitate the disease follow-up and monitoring of patients’ response to treatment is a common trend in cancer research. Results of our study demonstrate that the combination of microfluidic digital PCR with COLD-PCR increases the sensitivity of cf*BRAFV600E* testing in liquid biopsy samples from patients with papillary thyroid cancer. These results also suggest that the implementation of liquid biopsy in clinical practice can improve risk stratification and the delineation of an optimal therapeutic strategy in patients with PTC.

## Figures and Tables

**Figure 1 jcm-09-02481-f001:**
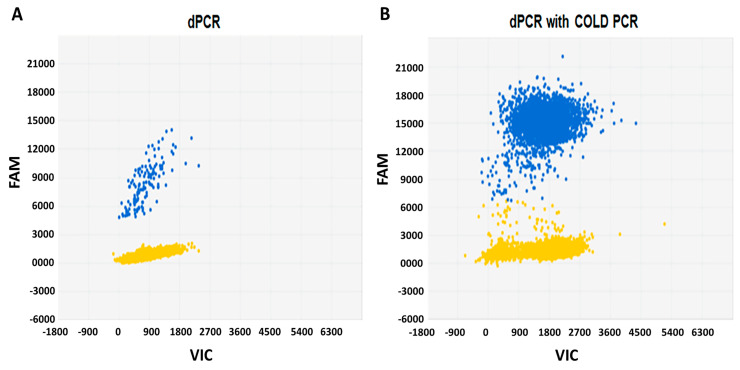
Detection of *BRAFV600E* by microfluidic digital PCR alone or in combination with COLD-PCR. Each panel represents a single experiment whereby DNA from BCPAP cells was segregated into individual wells and assessed for the presence of the mutant alleles using two different fluorophores (FAM and VIC). The signals from the FAM (blue) and VIC (red) dyes are plotted on the Y axis and the X axis, respectively. The yellow cluster represents the unamplified wells (negative calls). (**A**) Two-dimensional plots of microfluidic digital PCR reads out of 1 ng DNA extracted from BCPAP cells. Blue cluster represents the wells that were positive for the *BRAFV600E* mutation. (**B**) Results of dPCR analysis after the enrichment of *BRAFV600E* by COLD-PCR demonstrating 100-fold increase in mutant alleles (blue cluster). dPCR, digital Polymerase chain reaction; COLD, co-amplification at lower denaturation temperature; PCR, Polymerase chain reaction; FAM, fluorescein; VIC, 2′-chloro-7′phenyl-1,4-dichloro-6-carboxy-fluorescein.

**Table 1 jcm-09-02481-t001:** Clinicopathological features of the *BRAFV600*-positive papillary thyroid carcinomas.

Clinical Features	cf*BRAFV600E* in Plasma (24 Cases)	Wild-Type cf*BRAF* in Plasma (33 Cases)	*p*-Value
Age at the time of diagnosis (years)	45.7 ± 16.9	46.5 ± 16.4	0.92
Sex (male/female)	8/16	9/24	0.62
Tumor subtype			
Classical	16 (67%)	22 (67%)	0.87
Follicular variant	2 (8%)	4 (12%)	
Tall cell variant	6 (25%)	7 (21%)	
Tumor size (cm)	2.99 ± 1.9	1.99 ± 1.6	0.03
Multifocal growth	10 (42%)	24 (73%)	0.03
Gross extrathyroidal extension	9 (38%)	4 (12%)	0.02
Lymph node metastasis	16 (67%)	15 (45%)	0.27
Distant metastases	8 (33%)	4 (12%)	0.05
Pulmonary micrometastases	7 (29%)	3 (9%)	0.04

**Table 2 jcm-09-02481-t002:** Risk stratification of the patients with *BRAFV600*-positive papillary thyroid carcinomas.

Risk Stratification	*BRAFV600E* in Plasma *n* = 24 Cases	Wild-Type *BRAF* in Plasma *n* = 33 Cases	*p*-Value
Low risk	4 (21%)	15 (79%)	0.02
Intermediate risk	8 (36%)	14 (64%)	0.48
High risk	12 (75%)	4 (25%)	0.002

**Table 3 jcm-09-02481-t003:** Response to the treatment related to risk classification and cf*BRAFV600E* detection.

Risk Group	# Cases	*BRAF* Status	# Cases	Treatment Outcome			
				Excellent Response	Indeterminate Response	Biochemically Incomplete	Structurally Incomplete
Low	19	mutant	4	1	1	2	0
		wild-type	15	14	1	0	0
Intermediate	22	mutant	8	3	3	0	2
		wild-type	14	8	4	0	2
High	16	mutant	12	2	0	4	6
		wild-type	4	1	0	1	2

**Table 4 jcm-09-02481-t004:** The risk factors associated with non-excellent response to treatment (either biochemical or structurally incomplete or indeterminate response).

Variable	Odds Ratio (OR)	95% Confidence Intervals (CI)		*p* Value
		Lower	Upper	
Age at diagnosis	0.99	0.95	1.04	0.80
cf*BRAFV600E* detection	4.68	1.26	17.32	0.02
Risk stratification (intermediate vs. low)	2.81	0.62	12.75	0.18
Risk stratification (high vs. low)	9.33	1.539	56.618	0.01

OR, odds ratio; CI, confidence intervals.

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
