# Peer review of "Detection of BRAFV600E in Liquid Biopsy from Patients with Papillary Thyroid Cancer Is Associated with Tumor Aggressiveness and Response to Therapy"

_jcm, 2020, doi:10.3390/jcm9082481_

Round 1
Reviewer 1 Report
In this paper the Authors aim to assess the feasibility of the use of BRAFV600E in Liquid Biopsy from
Patients with Papillary Thyroid Cancer in order to identify tumor aggressiveness. It is a debated and novel topic. The number of cases is extremely limited but the method appears interesting. A comprehensive and extensive literature review of the NCBI database PubMed was also carried out. The article was well conducted and it is interesting in its fields. It is a well-structured paper, written in good English and the References are up dated.
Minor issues:
The use of an effective method in the early identification of thyroid tumor permit a minimal invasive surgery preventing some of the complication related to conventional surgery. In the “discussion” section I suggest to better analyze surgical complications of thyroidal surgical intervention. Therefore, the following paper should be considered:
“Calò PG, Medas F, Conzo G, et al. Intraoperative neuromonitoring in thyroid surgery: Is the two-staged thyroidectomy justified?. Int J Surg. 2017;41 Suppl 1:S13‐S20. doi:10.1016/j.ijsu.2017.02.001.”
“Gambardella C, Polistena A, Sanguinetti A, Patrone R, Napolitano S, Esposito D, Testa D, Marotta V, Faggiano A, Calò PG, Avenia N, Conzo G. Unintentional recurrent laryngeal nerve injuries following thyroidectomy: Is it the surgeon who pays the bill? Int J Surg. 2017 May;41 Suppl 1:S55-S59. doi: 10.1016/j.ijsu.2017.01.112. Review. PubMed PMID: 28506414.”
Author Response
Comments and Suggestions for Authors
In this paper the Authors aim to assess the feasibility of the use of BRAFV600E in Liquid Biopsy from Patients with Papillary Thyroid Cancer in order to identify tumor aggressiveness. It is a debated and novel topic. The number of cases is extremely limited but the method appears interesting. A comprehensive and extensive literature review of the NCBI database PubMed was also carried out. The article was well conducted and it is interesting in its fields. It is a well-structured paper, written in good English and the References are up dated.
Reply- Thank you for this feedback.
Minor issues:
The use of an effective method in the early identification of thyroid tumor permit a minimal invasive surgery preventing some of the complication related to conventional surgery. In the “discussion” section I suggest to better analyze surgical complications of thyroidal surgical intervention. Therefore, the following paper should be considered:
“Calò PG, Medas F, Conzo G, et al. Intraoperative neuromonitoring in thyroid surgery: Is the two-staged thyroidectomy justified?. Int J Surg. 2017;41 Suppl 1:S13‐S20. doi:10.1016/j.ijsu.2017.02.001.”
“Gambardella C, Polistena A, Sanguinetti A, Patrone R, Napolitano S, Esposito D, Testa D, Marotta V, Faggiano A, Calò PG, Avenia N, Conzo G. Unintentional recurrent laryngeal nerve injuries following thyroidectomy: Is it the surgeon who pays the bill? Int J Surg. 2017 May;41 Suppl 1:S55-S59. doi: 10.1016/j.ijsu.2017.01.112. Review. PubMed PMID: 28506414.”
Reply- We thank you for your consideration. We modified discussion and included aforementioned references (Page 9, Line 374).
“The use of an effective method in the early identification of thyroid tumor could permit a minimal invasive surgery preventing some of the complication related to conventional surgery [37,38]. Our data suggest the potential utility of liquid biopsy in guiding the surgical decision, however, additional studies addressing this specific question are needed.”
Reviewer 2 Report
In this study Kirk Jensen et al. compared the performance of microfluidic digital PCR alone and co-amplification at lower denaturation temperature (COLD) PCR in detecting the presence of BRAFV600E in cell-free DNA in plasma samples from patients with papillary thyroid cancer (PTC). They found a superiority of COLD-PCR and a correlation between cell-free BRAFV600E and some clinical features and higher risk tumors. This is a linear well conduct study. The Discussion and Introduction can be shortened.
Author Response
Comments and Suggestions for Authors
In this study Kirk Jensen et al. compared the performance of microfluidic digital PCR alone and co-amplification at lower denaturation temperature (COLD) PCR in detecting the presence of BRAFV600E in cell-free DNA in plasma samples from patients with papillary thyroid cancer (PTC). They found a superiority of COLD-PCR and a correlation between cell-free BRAFV600E and some clinical features and higher risk tumors. This is a linear well conduct study. The Discussion and Introduction can be shortened.
Reply- Thank you for this feedback. Some very minor changes were made in the Introduction but did not shorten the text. The introduction remains 1 page in length.
Due to redundancy and restating of portions of the Introduction in the Discussion section, 6 lines were deleted from line #316-324 (Page 8). This brings the Discussion section to two pages.
Reviewer 3 Report
In the study performed by Jensen et al. the authors examined the utility of “liquid biopsy” in 57 patients with PTC and BRAFV600E mutation. They hypothesized that detection of BRAFV600E in plasma samples from patients with BRAFV600E-positive primary PTC could be correlated with aggressiveness of TC.
The positivity of the work:
#they developed a protocol for analysis of cfBRAFV600E where microfluidic digital PCR was combined with COLD PCR, and the results suggest that the combination of microfluidic digital PCR with COLD PCR could be a suitable technique for assessment of “liquid biopsy” samples from patients with TC,
#what is more, the authors revealed that the presence of cfBRAFV600E was associated with the size of the primary tumor as well as with the presence of gross ETE,
#also pulmonary metastases were more frequently detected in patients with detectable cfBRAFV600E,
#they demonstrated that analysis of “liquid biopsy” can help in identification of a subset of PTCs with more aggressive features even in the group of low-risk cancer patients,
#they demonstrated that detection of cfBRFV600E in plasma is an independent factor predicting response to therapy,
#what is more circulating BRAFV600E may provide an additional tool to monitor PTC patients during treatment,
#clear statistical analysis,
#clear design,
#clear language.
Some limitations of the work:
#retrospective design,
#small number of analyzed samples (patients),
#the usual plasma test named a little cleverly/misleadingly as “liquid biopsy” – why?,
#lack of the whole name of the abbreviations as “TERT” and “LT4” when first used in the study – please clarify,
#using not professional/scientific term like “aggressive histology” – please replace.
In my opinion, the work is worthy of further evaluation.
Author Response
Comments and Suggestions for Authors
In the study performed by Jensen et al. the authors examined the utility of “liquid biopsy” in 57 patients with PTC and BRAFV600E mutation. They hypothesized that detection of BRAFV600E in plasma samples from patients with BRAFV600E-positive primary PTC could be correlated with aggressiveness of TC.
The positivity of the work:
#they developed a protocol for analysis of cfBRAFV600E where microfluidic digital PCR was combined with COLD PCR, and the results suggest that the combination of microfluidic digital PCR with COLD PCR could be a suitable technique for assessment of “liquid biopsy” samples from patients with TC,
#what is more, the authors revealed that the presence of cfBRAFV600E was associated with the size of the primary tumor as well as with the presence of gross ETE,
#also pulmonary metastases were more frequently detected in patients with detectable cfBRAFV600E,
#they demonstrated that analysis of “liquid biopsy” can help in identification of a subset of PTCs with more aggressive features even in the group of low-risk cancer patients,
#they demonstrated that detection of cfBRFV600E in plasma is an independent factor predicting response to therapy,
#what is more circulating BRAFV600E may provide an additional tool to monitor PTC patients during treatment,
#clear statistical analysis,
#clear design,
#clear language.
Some limitations of the work:
#retrospective design,
#small number of analyzed samples (patients),
#the usual plasma test named a little cleverly/misleadingly as “liquid biopsy” – why?,
Reply- Thank you for the feedback. The term ‘liquid biopsy’ has become commonplace to describe the testing of mutation status and other novel biomarkers from blood or urine. Indeed, there are over 1,000 PubMed cited articles in just the past year that refer to the use of liquid biopsy for the detection of these various biomarkers.
#lack of the whole name of the abbreviations as “TERT” and “LT4” when first used in the study – please clarify,
Reply- Thank you for recognizing these shortcomings and bringing it to our attention. Telomerase reverse transcriptase (TERT) is added to the text at line #55-56 (Page 2). Levothyroxine (LT4) is added to the text at line #120 (Page 3).
#using not professional/scientific term like “aggressive histology” – please replace.
Reply- Thank you for the comment. The phrases used in the paragraph starting from Line #107-110 (Page 3) were altered to give a clearer indication of the presentation of the tumor samples and the basis for their classification into low-, intermediate- and high-risk groups.